# A Narrative Review of the Molecular Epidemiology and Laboratory Surveillance of Vaccine Preventable Bacterial Meningitis Agents: *Streptococcus pneumoniae*, *Neisseria meningitidis*, *Haemophilus influenzae* and *Streptococcus agalactiae*

**DOI:** 10.3390/microorganisms9020449

**Published:** 2021-02-22

**Authors:** Raymond S. W. Tsang

**Affiliations:** Laboratory for Vaccine Preventable Bacterial Diseases, National Microbiology Laboratory, Public Health Agency of Canada, 1015 Arlington Street, Winnipeg, MB R3E 3R2, Canada; raymond.tsang@canada.ca; Tel.: +1-204-789-6020

**Keywords:** bacterial meningitis, *S. pneumoniae*, *N. meningitidis*, *H. influenzae*, *S. agalactiae*, conjugate vaccines, post-vaccine surveillance

## Abstract

This narrative review describes the public health importance of four most common bacterial meningitis agents, *Streptococcus pneumoniae*, *Neisseria meningitidis*, *Haemophilus influenzae*, and *S. agalactiae* (group B *Streptococcus*). Three of them are strict human pathogens that normally colonize the nasopharynx and may invade the blood stream to cause systemic infections and meningitis. *S. agalactiae* colonizes the genito-gastrointestinal tract and is an important meningitis agent in newborns, but also causes invasive infections in infants or adults. These four bacteria have polysaccharide capsules that protect them against the host complement defense. Currently licensed conjugate vaccines (against *S. pneumoniae*, *H. influenza*, and *N. meningitidis* only but not *S. agalactiae*) can induce protective serum antibodies in infants as young as two months old offering protection to the most vulnerable groups, and the ability to eliminate carriage of homologous serotype strains in vaccinated subjects lending further protection to those not vaccinated through herd immunity. However, the serotype-specific nature of these vaccines have driven the bacteria to adapt by mechanisms that affect the capsule antigens through either capsule switching or capsule replacement in addition to the possibility of unmasking of strains or serotypes not covered by the vaccines. The post-vaccine molecular epidemiology of vaccine-preventable bacterial meningitis is discussed based on findings obtained with newer genomic laboratory surveillance methods.

## 1. Introduction

Pyogenic bacterial meningitis is a life threatening condition that can progress rapidly leading to death. When the disease happens in infants, children, and young adults, it may instill fear due to the contagious and potentially deadly nature of the disease especially in outbreak situation. The three most common causes of acute bacterial meningitis are *Streptococcus pneumoniae*, *Neisseria meningitidis*, and *Haemophilus influenzae* [1]. This group of bacterial meningitis agents can cause disease in all ages of life from newborn to the elderly. The global burden of meningitis disease in 2016 was estimated to be 2.82 million cases, and 318,400 deaths were attributed to meningitis. The three most common pathogens (*S. pneumoniae*, *N. meningitidis*, and *H. influenzae*) were responsible for 55.7% and 57.2% of the meningitis cases and deaths, respectively [2]. Besides meningitis, *S. pneumoniae*, *H. influenzae*, and *N. meningitidis* can cause other forms of invasive diseases such as bacteremic pneumonia, septicemia, septic arthritis, pericarditis, etc. The risk of developing a major (such as hearing loss, seizures, motor deficit, cognitive impairment, hydrocephalus, and visual disturbance) or a minor (learning difficulties, language impairment, developmental delay) sequela from bacterial meningitis was estimated to be 12.8% and 8.6%, respectively [3]. Meningitis caused by *S. pneumoniae* carried the highest risk with a major sequela (24.7%), followed by *H. influenzae* (9.5%) and *N. meningitidis* (7.2%) [3]. Using meningococcal disease (which carries the lowest risk of developing a major sequela) as an example, the cost to care for a case who developed a major sequela was estimated to be £160,000 (US$214,096) to £200,000 (US$267,620) for the first year alone; and the corresponding figure over the lifetime of a case may be as high as £590,000 (US$789,479) to £1,090,000 (US$1,458,529) [4]. Since the incidence of meningitis and the risk of developing sequela are much higher in low- and middle-income countries, and the resources to care for those meningitis patients who develop severe sequela are often lacking in these countries, vaccines are probably the most cost-effective strategy for the control and potentially elimination of this devastating and fearful disease.

Although a number of other bacterial agents can cause meningitis, such as *Listeria monocytogenes*, *Escherichia coli,* and other enteric bacteria, group B *Streptococcus* (*S. agalactiae*) is gaining attention as a frequent cause of either early or late onset of invasive diseases such as pneumonia, sepsis, or meningitis in the newborn [5,6] as well as various forms of invasive diseases in pregnant women and non-pregnant adults [5,7]. The World Health Organization (WHO) has also identified group B *Streptococcus* together with *S. pneumoniae*, *N. meningitidis*, and *H. influenzae* as the four major bacterial meningitis agents to be included in its work plan and global vision to defeat meningitis by 2030 [8].

Capsule-based protein-conjugate vaccines that target the major serogroups of *N. meningitidis* and serotypes of *H. influenzae* and *S. pneumoniae* causing invasive diseases are now available and implemented in vaccination programs in many countries [9,10,11]. As a result, the epidemiology of bacterial meningitis has changed with the number of cases caused by strains covered by the vaccine decreased dramatically but at the same time disease due to serogroups or serotypes of the pathogens not included in the vaccine has emerged [12]. Since disease surveillance has been described by the WHO as one of the five major pillars on the road map to defeat meningitis [8], the objectives of this report are to describe (i) features of *S. pneumoniae*, *N. meningitidis*, *H. influenzae,* and *S. agalactiae* that may have implications for vaccination and surveillance; (ii) currently licensed vaccines against *S. pneumoniae*, *N. meningitidis*, and *H. influenzae*; (iii) changes in the epidemiology of invasive diseases caused by these three pathogens; (iv) traditional and newer laboratory surveillance methods; and (v) how lessons learned from surveillance of the three most common bacterial meningitis agents can inform the pre- and post-vaccine licensure surveillance of invasive group B *Streptococcus* (GBS) disease when capsule polysaccharide conjugate vaccines against GBS have been developed and are in clinical trials [5,13].

## 2. Characteristics of *S. pneumoniae*, *N. meningitidis*, *H. influenzae*, and *S. agalactiae* Important for Vaccination and Surveillance

*S. pneumoniae*, *N. meningitidis*, and *H. influenzae* are respiratory pathogens that normally colonize the human respiratory tract where they serve as a reservoir of infection [14,15,16]. Another common characteristic of these three invasive bacterial agents is the polysaccharide capsules on their cell surface, which serve as serotyping antigens. The serotypes are traditionally identified by anti-capsular antibodies using agglutination methods (or the Quellung reaction for *S. pneumoniae*). The capsules also serve as protective antigens shielding the bacteria from the human host defense like phagocytosis and complement activation [17,18]. As the protective antigen, vaccines based on the capsule have been developed to target the most common serotypes of *H. influenzae*, *N. meningitidis*, and *S. pneumoniae* causing invasive infections [9,10,11]. Another feature that makes these bacteria successful pathogens is the plasticity of their genome and their recombinant nature [19,20,21].

Unlike *S. pneumoniae*, *N. meningitidis*, and *H. influenzae*, *S. agalactiae* colonizes the human genito-gastrointestinal tract. Not only does it cause meningitis in the newborn and various forms of invasive diseases in infants and adults, *S. agalactiae* is also known to cause disease in cattle [22,23] and may have the potential to transmit to human as a zoonotic pathogen [24]. Similar to *S. pneumoniae*, *N. meningitidis*, and *H. influenzae*, *S. agalactiae* also has a surface polysaccharide capsule that acts as virulence factor and protective antigen [5,13]. Its genome is also prone to participate in recombination events [25].

## 3. Currently Licensed Vaccines for Control of Bacterial Meningitis

Currently there are 6 serotypes of *H. influenzae* recognized [26], 10 serotypes of *S. agalactiae* [5,13], 12 serogroups for *N. meningitidis* [27], and 100 serotypes for *S. pneumoniae* [28,29] (Table 1). Non-encapsulated strains also exist in all three species, and are termed non-typeable (for *H. influenzae* and *S. pneumoniae*) or non-groupable (for *N. meningitidis*). Currently licensed vaccines to control some strains of *S. pneumoniae*, *N. meningitidis*, and *H. influenzae* are listed in Table 2, while vaccines for protection against *S. agalactiae* are not licensed yet but are in advanced stages of clinical trials for maternal immunization [30,31].

The first bacterial meningitis vaccine developed was solely polysaccharide-based vaccine against serotype b *H. influenzae* (Hib) but it was soon discovered that plain polysaccharide vaccines are T-cell-independent antigens and do not induce protective antibodies in infants less than two years old [32], the most vulnerable age group for developing meningitis and invasive disease [33,34]. Coupling of the capsular polysaccharide to a protein carrier converts the vaccine to a T-cell dependent antigen that induces protective antibodies in infants as young as two months of age [10]. Another characteristic of the capsular polysaccharide vaccines is they are serotype-specific and offer protection against infection by the homologous serotype and do not offer protection against heterologous serotypes. Besides preventing invasive infections, the conjugate vaccines also reduce or eliminate respiratory carriage of and hence offer herd immunity to the larger community for the serotypes of these pathogens included in the vaccines [35,36,37]. The fact that conjugate vaccines are serotype-specific and can eliminate nasopharyngeal carriage of the homologous serotypes means their protective coverage is limited to the serotypes included in the vaccine and they can also alter the bacterial flora in the nasopharynx of vaccinated subjects.

The choice of which serotypes or serogroups to be included in the vaccines are based on the fact that not all serotypes or serogroups are equally virulent nor have the same prevalence in causing invasive diseases. For example, animal infection with isogenic mutants of *H. influenzae* that expressed different capsule serotype antigens has shown that serotype b is the most virulent, followed by serotype a [38]. In addition, most *N. meningitidis* isolates recovered from normally sterile body sites of invasive meningococcal disease (IMD) patients belong to six of the 12 recognized serogroups (A, B, C, W, X, and Y) [39,40]. Before the introduction of pneumococcal conjugate vaccines (PCVs), 10 serotypes (1, 4, 5, 6A 6B, 14, 18C, 19A, 19F, and 23F) were responsible for at least 50% of all invasive pneumococcal disease isolates from six different parts of the world; and in one region, they were responsible for over 80% of their invasive pneumococci [41].

Even before vaccine introduction, temporal and geographical variations in the serogroups of *N. meningitidis* responsible for IMD is well documented [39,40]. Differences in the serotypes involved in invasive pneumococcal disease (IPD) have also been reported from different parts of the world [42,43]. Before Hib conjugate vaccines were introduced, most invasive *H. influenzae* diseases were caused by Hib [33,34].

## 4. Effects of Vaccine Pressure, and Immune and/or Antibiotic Selection

Since currently licensed conjugate vaccines against *S. pneumoniae*, *N. meningitidis*, and *H. influenzae* do not offer universal coverage for all the serotypes or serogroups, immune pressure and selection against these pathogens can be expected to happen in their natural habitat as well as in the serotypes and serogroups that will cause invasive disease in the post-vaccine period.

In the presence of vaccine pressure, these bacterial pathogens may evolve to adapt by mainly two mechanisms that affect their capsule antigens (the vaccine targets): Capsule or serotype switching, and capsule or serotype replacement. Capsule switching involves two strains of a species exchanging their capsule polysaccharide synthesis (*cps*) genes resulting in a swap of their capsule antigens. For example, as shown in Figure 1a, a strain of genetic linage 1 and with a vaccine type capsule (depicted in green) exchanges its *cps* genes with a strain of genetic lineage 2 and with a non-vaccine capsule type (depicted in red). The end result will be the genetic linage 1 strain now carries *cps* genes for non-vaccine capsule type and expresses the non-vaccine capsule (red); while conversely the genetic linage 2 strain now expresses vaccine type capsule (green). Both *S. pneumoniae* and *N. meningitidis* have been reported to have capsule switching occurring spontaneously in the absence of vaccine pressure or, i.e., such capsule switching events have been reported prior to conjugate vaccine introduction [44,45]. Capsule switched strains can also be selected for by vaccine induced immune pressure and/or by wide spread antibiotic use if the capsule switched strain carries antibiotic resistance genes [41]. After capsule switching, the recipient strain will retain its original genetic background (usually determined by multi-locus sequence typing) [46] but expresses a different, e.g., non-vaccine type of capsule. Frequent capsule switch in *N. meningitidis* from serogroup C to serogroup B, if it happens in a hypervirulent clone like ST-11, may be problematic since there are no capsule-based serogroup B meningococcal vaccines and protein-based meningococcal vaccines against serogroup B may not provide universal coverage against all serogroup B strains [47].

Capsule replacement happens when strains with vaccine capsule types that used to inhabit the nasopharynx have been eliminated by the conjugate vaccines and are now being replaced by strains expressing the non-vaccine capsule types. As a result, strains of non-vaccine capsule types may increase in prevalence and eventually cause disease. Strains with vaccine capsule type (depicted as green) and non-vaccine capsule type (depicted as red) can co-exist prior to the use of conjugate vaccines and the two capsule types may be of the different (Figure 1b) or the same (Figure 1c) genetic lineage. After selection by vaccine pressure, only strains of the non-vaccine capsule type (red) remain and expand to fill the void that used to be occupied by strains of the vaccine type (Figure 1b). If strains of the same genetic lineage that expressed both vaccine and non-vaccine capsule types exist prior to vaccine use, then strains of the same genetic lineage still persist after vaccine use but they only carry the non-vaccine capsule type (Figure 1c). The phenomenon illustrated in Figure 1b,c is sometimes referred to as “unmasking” versus true replacement when only the vaccine capsule type exists in the nasopharynx before vaccine use, and after vaccine use, the vaccine capsule type strain is removed and the void in the nasopharynx is being replaced by a non-vaccine capsule type strain. Strains of non-vaccine capsule types can also be further selected by widespread antibiotic use if they carry the corresponding antibiotic resistance genes [41].

## 5. Molecular Epidemiology of Invasive Pneumococcal Disease (IPD), Invasive Meningococcal Disease (IMD), and Invasive *H. influenzae* Disease in the Post Conjugate Vaccine Era

### 5.1. IPD in the Post PCV Era

Although PCVs are very effective in reducing the burden of IPD caused by vaccine serotypes in many countries, IPD due to non-vaccine serotypes is still a concern. The emergence of IPD due to non-vaccine serotypes may be due to either unmasking effect when vaccine serotypes are removed from their natural habitat of the human nasopharynx, allowing non-vaccine serotypes (which already exist) to expand and occupy the nasopharynx; or by replacement due to the non-vaccine serotypes which do not exist in the nasopharynx prior to PCV introduction but emerge to fill the void in the nasopharynx left behind by the vaccine type [50]. For example, prior to PCV7 introduction, serotypes 19A (a non-PCV7 vaccine serotype) existed at a level of about 7.5% in 2001 and increased to 16% in 2007 after PCV7 was introduced in 2000, before declining to about 3% in 2014 after PCV13 was introduced in 2010 [51]. The other mechanism responsible for the emergence of non-vaccine serotypes is genetic recombination between strains leading to capsule switching. This was illustrated in the emergence of some serotype 19A strains after PCV7 introduction by a genetic recombination between a vaccine covered serotype 4, sequence type (ST)-595 recipient strain with a donor strain of non-vaccine serotype 19 ST-199, providing the recipient ST-595 strain with the non-vaccine capsule serotype 19A [52]. Another study also demonstrated that serotype 19A may again escape the PCV13 selection by a further switch to serotype 15B [53]. Although in this later study the genetic recombination occurred in strains from pre-PCV7 period; nevertheless the mechanism of genetic recombination with a non-vaccine capsule type is present in pneumococci. The pneumococcal capsule locus is a hotspot for mutation including exhibiting a higher rate of genetic recombination compared to the rest of the pneumococcal genome [54]. However, pneumococcal capsule locus recombination that leads to capsule serotype switch does not appear to be random. For example, capsule switch between strains within a serogroup occurred more often than serotype switch involving strains between different serogroups [55]. Since many factors may govern the pneumococcal population structure and the associated serotypes, some have suggested the existence of epistatic factor contributing to the dynamic of the pneumococcal capsule genetics [55,56].

Another mechanism may explain the persistence of some vaccine serotype in the post PCV period. For example, serotype 3 (included in the PCV13 vaccine) persisted in the nasopharyngeal samples as well as in specimens from IPD patients despite PCV13 usage [51,57,58]. Genome sequencing of serotype 3 isolates obtained prior to and after introduction of PCV13 showed a different clade of serotype 3 has emerged in the post PCV13 period despite the fact that both pre- and post-PCV13 isolates were typed by MLST to belong to the same ST-180 clonal complex (CC). However, the new clade has been shown to have sub-capsular protein antigen changes, which could explain strains of the new clade have adapted to exist despite the presence of immunity induced by PCV13 [57].

Regardless of the vaccine escape mechanism, various non-PCV serotypes have emerged in places where PCV immunization programs have been implemented reflecting geographical differences in serotype prevalence and distribution [59]. Non-PCV serotypes like serotype 2, 8, 10A, 11A, 12F, 15A, 15B/C. 16F, 22F, 24F, 33F, and 35B/D, have been described as causes of IPD [41,59,60,61,62,63]. To deal with this increase in non-PCV serotypes, 15-valent and 20-valent PCVs have been developed and are now in early clinical trials [64,65]. However, in the post PCV era, predominance by a single or a few serotypes as causes of IPD was not observed. Instead, increase serotype diversity of invasive pneumococci recovered from IPD cases has been observed [66], which may challenge the usefulness of increasing the valency of PCVs. Two editorials in 2007, “Invasive pneumococcal disease, the target is moving” [67] and “Serotype replacement in invasive pneumococcal disease: where do we go from here?” [68] appear to be just as relevant today after two decades of PCV use. Indeed, expert comments in 2021 still wrestle with the changing epidemiology of IPD due to shifting serotypes, and identify continuous surveillance as an important function in the control of IPD [69,70]. Ideally, a pan-pneumococcal universal vaccine would solve the problem of chasing after the emergence of non-vaccine serotypes as causes of IPD.

### 5.2. IMD in the Post Conjugate Vaccine Era

In the US, quadrivalent (A, C, W, Y) meningococcal conjugate vaccine was licensed in 2005 and recommended for the 11 to 18 years age group [71]. In the post-quadrivalent conjugate vaccine period of 2006-2010, no capsule or serogroup replacement was detected [72]. In the Canadian province of Quebec, outbreak due to a serogroup B strain of ST-269 appeared in 2013 [73] after two rounds of province wide vaccination against serogroup C meningococci (MenC) (first with plain polysaccharide vaccine in 1992-1993 and then with the MenC-conjugate vaccine in 2011) for control of outbreaks due to the hyper-virulent strain of ET-15 (ST-11) [74,75].

In Europe, serogroup B *N. meningitidis* was responsible for most IMD (73.6% in 2011) while an increase in serogroup Y IMD has been reported in a number of European countries [40]. Beginning in 2013, an increase in IMD due to serogroup W meningococci (MenW) has been reported from across Europe with both incidence rates of disease and the proportion of IMD isolates due to serogroup W showing yearly increase [76]. This increase in MenW disease in Europe was due to the introduction of a new ST-11 strain (different from the Hajj strain, which emerged during the 2000 Hajj pilgrimage in Saudi Arabia) from South America into the UK with further diversification to the 2013 UK strain, which spread through Europe [76,77]. This new MenW strain has also been reported to cause an increase in IMD in both Australia and Canada [78,79]. Expansion of a penicillin-resistant MenW ST-11 clone has also been described [80].

Before introduction of the monovalent meningococcal serogroup A conjugate vaccine, MenAfriVac, in 2010 [81], most meningococcal epidemics in Africa were mostly caused by serogroup A *N. meningitidis* (MenA). However, a serogroup X meningococcus (MenX) epidemic was reported from southwest Niger in 2004-2006 [82], and subsequently MenX outbreaks had occurred in Burkina Faso, Niger, Togo, and Uganda [83]. The MenX outbreak strain has been characterized as ST-181 CC with high experimental animal pathogenicity [84]. After introduction of the MenAfriVac, epidemics due to serogroups C, W, and X meningococci have been reported in the African meningitis belt countries. [85,86]. The MenC strain appeared to be a new strain typed as ST-10217, which has been shown to have arisen from a nongroupable strain recovered from a healthy carrier in Burkina Faso in 2012 prior to the emergence of the MenC ST-10217 and the MenC outbreak in 2013 [87]. The MenW strain causing outbreaks in Africa has been studied, and it appeared to be related to, and to have diversified from the 2000 Hajj strain [88].

Longitudinal carriage studies have been carried out in Africa to understand the epidemic nature of meningococcal meningitis in recent years both before and after the introduction of MenAfriVac in 2010. In the study carried out in one district of northern Ghana over the period of 1998 to 2005 before MenAfriVac was introduced, it was found that the colonized meningococcal population changed with time and matched temporally with the strain causing epidemics in the region [89]. Three successive waves of colonized meningococci were observed with ST-5 MenA, followed by ST-751 MenX, and ST-7 MenA. In the study to assess the effect of immunization with MenAfriVac, the carriage study has shown that the vaccine was both effective in control of MenA disease and in elimination of MenA from the respiratory tract of healthy carriers up to six or seven years after vaccine introduction. Like the other longitudinal carriage study in Ghana, a small percentage of oropharyngeal samples contained MenW of ST-11 CC (0.48%) and MenC of ST-10217 CC (0.10%) [90]. These studies certainly pointed to the importance of meningococci in healthy carriers as contributors of infection and potential sources of epidemics; and that conjugate vaccination may further change the population of meningococci in the normal habitat of the human upper respiratory tract.

### 5.3. Invasive H. influenzae Disease in the Post Hib Conjugate Vaccine Era

Following introduction of the Hib conjugate vaccine in the early 1990s, the epidemiology of invasive *H. influenzae* disease in those countries with Hib vaccination programs have changed substantially in the past three decades. Non-typeable or non-encapsulated *H. influenzae* (NT-Hi) is now the most frequent cause of invasive *H. influenzae* disease worldwide [91]. In Europe, during the period of 2007 to 2014, NT-Hi was the most common type identified but 74.1% of their invasive encapsulated *H. influenzae* were typed as serotype f, followed by serotype e (21.4%) [92]. In contrast, in the U.S., although NT-Hi is also the most common cause of invasive *H. influenzae* disease, the incidence of serotype a invasive *H. influenzae* disease has increased by 13% annually during the period of 2002 to 2015 [93].The incidence of invasive disease caused by NT-Hi has increased by 3% annually while incidence of invasive *H. influenzae* disease due to other serotypes was either stable or decreasing. The global presence of serotype a *H. influenzae* (Hia) has been documented [94,95], and the severity of invasive Hia disease has been described [96,97,98] which called for a Hia vaccine development [99].

Genetic analysis of Hia has revealed a population biology very similar to Hib, i.e., (a) with two phylogenetic populations similar to the clonal divisions I and II descried for Hib; and (b) with most invasive Hia isolates clustered together in a phylogenetic population (named clonal division I as for the majority of invasive Hib strains), represented by isolates typed by MLST as ST-23 and many STs related to ST-23 as single, double, or triple locus variants [100,101]. Another clone within this larger genetic population of clonal division I and identified by MLST as ST-4 has been reported in Brazil to be associated with more severe disease and higher case fatality rate [102]. In contrast to clonal division I Hia, clonal division II Hia is rarely isolated from invasive disease cases in Canada [101] and has not been found associated with invasive disease in Alaska [100]. However, clonal division II Hia identified by MLST as ST-62 has been found in 75% (21/28) of the Hia invasive disease case isolates from children < 18 years old in Utah, United States [103]. This may suggest unique geographical distribution of Hia genotypes.

To understand the emergence of NT-Hi as a cause of invasive disease in the post Hib conjugate vaccine era, comparative genome studies have revealed that NT-Hi showed much higher genetic diversity when compared to Hib or other serotypes that have been regarded as more genetically conserved or clonal [104,105]. Non-encapsulated *S. pneumoniaae* have also been reported to have higher genetic diversity probably as a result of higher rates of genetic recombination as the capsule may serve as a barrier for foreign DNA uptake [21,106]. The higher genetic diversity of non-encapsulated *H. influenzae* may offer better adaptation to the host, e.g., by evading host immunity.

## 6. Laboratory Surveillance of *S. pneumoniae*, *N. meningitidis*, and *H. influenzae*

It is with this background of a changing microbial ecology in the nasopharynx of the human host as the bacterial pathogens are adapting to the vaccine pressure that laboratory surveillance of invasive bacterial meningitis pathogens are becoming increasingly important as well as challenging. The molecular typing methods for outbreak detection and surveillance of IMD, IPD, and invasive *H. influenzae* disease have been reviewed a decade ago with a focus on DNA sequencing methods [107]. They can be briefly summarized below as:

(1) Serogrouping and serotyping by the conventional method of using antisera to detect the capsular antigens, through either bacterial agglutination or Quelling reactions, has the value of detecting expression of the capsule antigens albeit the method may sometimes be inaccurate [108]. Molecular method like PCR has been introduced to improve the detection and identification of serotypes, including that for *S. agalactiae* [109,110,111,112].

(2) Clonal analysis by multilocus sequence typing (MLST) is available for *S. pneumoniae*, *N. meningitidis,* and *H. influenzae* [46,113,114]. Strains are typed as STs and related STs are grouped together to form a CC.

(3) Target gene sequencing for fine typing:

The following genes have been proposed for typing of meningococci: *fetA,* which encodes an iron-regulated outer membrane protein; *porA*, which encodes the class 1 outer membrane protein PorA; *porB*, which encodes the class 2/3 outer membrane protein PorB [107]; and the newer protein-based MenB vaccine target genes, *fHbp*, *nhba*, and *nadA*, which encode for factor H binding protein, Neisseria heparin binding antigen, and the Neisseria adhesion A, respectively [115,116].

For *S. pneumoniae*, the *ply* and *lytA* genes, which encode for the pneumolysin and the autolysin, respectively; as well as the *pspA* gene, which encodes for the pneumococcal surface protein A, have been proposed as targets for potential typing purposes [107]. PspA has been reported to be associated with virulence and invasiveness of pneumococci [117]. Other gene markers associated with virulence have been suggested, including *pspC*, which encodes for the pneumococcal choline-binding protein C (PspC), for association with invasiveness of strains [117]. The *slaA* gene, which encodes for phospholipase A2, and four contiguous genes, one of which predicted as *pblB* that encodes a prophage tail protein, were either associated with the clinical disease of meningitis, or 30-day mortality rate, respectively [118].

The following genes have been proposed for typing of *H. influenzae*: *ompP2*, *ompP5*, *hmw1*, and *hmw2* [107]. The *omp2* and *omp5* genes encode for two different outer membrane proteins, a porin and a OmpA family protein, respectively. The *hmw1* and *hmw2* encode for HMW1 and HMW2, which are surface adhesion proteins (HMW stands for high molecular weight). A number of potential vaccine candidates have also been identified and they may have potential as further typing targets [119].

(4) Antibiotic susceptibility profile: Antibiogram can serve as a typing tool but more usefully in direct patient care as well as for surveillance purpose. Testing can be done by the disk diffusion method or quantitatively by the dilution assays (broth or agar dilution methods). Guidelines for the testing methods including the culture media, classes of antibiotics to be tested, as well as the interpretation of results have been published by both The European Committee on Antimicrobial Susceptibility Testing (EUCAST) [120] and the Clinical Laboratory Standards Institute’s (CLSI’s) Subcommittee on Antimicrobial Susceptibility Testing (AST) [121].

Besides the phenotypic methods, genetic prediction of antibiotic susceptibility has also been described by Harrison et al. [107]. Of the genes associated with decreased susceptibility or resistance to different types of antibiotics, the penicillin binding protein genes of *S. pneumoniae* that determines susceptibility towards penicillin are of special interest. The *cps* locus of *S. pneumoniae* is flanked by two of the penicillin binding protein genes, *pbp1a* and *pbp2x*; their juxtaposition sometimes allow capsule switching and transfer of the penicillin resistant genes to occur in a single recombination event.

## 7. Whole Genome Sequencing (WGS) for Molecular Epidemiology and Genomic Surveillance of Vaccine-Preventable Bacterial Meningitis Agents

For quite some years, MLST has been proven useful to classify isolates into clonal types and it has been applied to identify hypervirulent clones [122,123] and capsule switching events between serogroups or serotypes [44,45]. However, intra-clonal variations have been described, which may have implications in our understanding of the changing epidemiology of these vaccine-preventable diseases [57,117,124].

With the first bacterial genome sequenced and published in 1995 [125], there has been a very rapid development over the last two decades in sequencing technologies that include cost reduction as well as web-based bioinformatics platforms and pipelines to assemble and analyze genome sequences. As such, genome sequencing has now become a standard laboratory tool to study microbes. Many of our current understanding of the molecular epidemiology of a number of infectious diseases, including the common bacterial meningitis agents of *S. pneumoniae*, *N. meningitidis*, *H. influenzae,* and *S, agalactiae* are based on data obtained by whole-genome sequencing projects.

However, to make WGS a routine laboratory tool for the global surveillance of vaccine-preventable bacterial meningitis, additional work may still be required on standardization and harmonization of methodology, data analysis and nomenclature on top of the issues of data ownership, and confidentiality. A global partnership to study the genomes of *S. pneumoniae* has published an international definition of pneumococcal lineage [126]. For *N. meninigitidis*, nomenclature is currently based on historical convention [127] and an international committee is responsible for naming clonal complexes (https://pubmlst.org/organisms/neisseria-spp/further-info (accessed on 15 December 2020)). Similar development for *H. influenzae* appears to be lacking for now. Traditional analysis of the population biology of encapsulated *H. influenzae* divided them into two clonal divisions [128], and WGS analysis of the recently emerged serotype a *H. influenzae* also revealed two populations [101] like the two clonal divisions described using multilocus enzyme electrophoresis of Hib [128]. However, the definition of lineages of non-typeable *H. influenzae* may need further study and discussion because their genetic background appear to be much more diverse than the encapsulated or serotypeable strains of *H. influenzae* [129].

WGS data can be used to predict results obtained by the traditional surveillance methods. Use of WGS to predict serotype of *S. pneumoniae* [130] and *H. influenzae* [131] as well as serogroup of *N. meningitidis* [132] have been described. A platform that uses WGS data for determination of MLST ST and clonal analysis has also been developed [133]. Use of WGS data to identify genetic typing markers and virulence factors has also been published [105,118]. Pipelines to apply WGS to predict antibiotic susceptibility of bacterial pathogens have been developed [134,135]. Improved sequencing technology has allowed direct non-culture genome sequencing from clinical specimens to identify the cause of culture negative fulminant fever [136]. This metagenomics approach has been applied to investigate a meningococcal outbreak in Liberia and the genome data identified the outbreak strain as identical to the unique serogroup C meningococcal strain causing outbreaks in West Africa [137]. The experience gained from WGS studies of *S. pneumoniae*, *N. meningitidis*, and to a lesser extent *H. influenzae*, would help to inform and prepare for the pre- and post-vaccine introduction surveillance of *S. agalactiae*. The platforms built for *S. pneumoniae*, *N. meningitidis*, and *H. influenzae* will likely shorten the deployment of these technologies to study *S. agalactiae*.

To enable a whole genome nucleotide sequence-based surveillance tool to complement conjugate vaccines in the global effort to defeat meningitis, a WHO-led partnership called Global Meningitis Genome Partnership (GMGP) was formed to coordinate, assist, and develop guidelines for using WGS data to identify and track the global epidemiology of common bacterial meningitis agents of *S. pneumoniae*, *N. meningitidis*, *H. influenzae*, and *S. agalactiae* [138]. This collaborative approach has the potential of building synergy between the international partners to achieve the goal of defeating vaccine-preventable bacterial meningitis by 2030.

## 8. Chemoprophylaxis, Corticosteroids, and Experimental Immune Modulating Approaches for Prevention and as Adjuvant Therapeutic Agents of Bacterial Meningitis

Although vaccines remain the primary tool to offer active protection against infections, chemoprophylaxis can prevent secondary cases by offering protection to close contacts and household members of index cases. Chemoprophylaxis can also offer protection to those immunized subjects before adequate level of adaptive immunity can be developed. Guidelines that define household members and close contacts of index cases of Hib and IMD as well as the choice and dosage of prophylactic antibiotics have been published [139,140]. For those requiring chemoprophylaxis to prevent IMD, a single dose of ciprofloxacin is recommended, or rifampicin given twice daily for two days as an alternative. Other prophylactic antibiotics may include ceftriaxone, cefixime, and azithromycin. IMD patients treated with benzylpenicillin (which may not eliminate pharyngeal meningococci) are recommended to receive chemoprophylaxis that can eliminate nasopharyngeal carriage of meningococci before hospital discharge to prevent potential transmission to household members. Rifampicin once a day for four days or ciprofloxacin twice a day for five days are recommended prophylactic antibiotics for contacts of index cases of Hib. Other effective antibiotics may include ceftriaxone and azithromycin. Chemoprophylaxis is generally not recommended for close contacts of IPD patients. However, children with increased risk of IPD such as those with asplenia or sickle cell disease should receive daily prophylaxis with oral penicillin [141]. Public Health England also has guidelines of infection control, vaccination, and chemoprophylaxis (with rifampicin, penicillin, or azithromycin) for high risk individuals living in closed settings when outbreak or cluster of severe pneumococcal disease occur [142]. To prevent early onset of GBS in neonates, pregnant women should be offered screening for GBS and intrapartum antibiotic prophylaxis in indicated situations [143]. Besides chemoprophylaxis, immunization with the recommended vaccines for IMD, Hib, and IPD should be the primary tool for prevention of these vaccine preventable diseases.

Early treatment with dexamethasone reduced mortality and improved the outcome of adult patients with acute meningitis [144]. However, in a Cochrane review to study corticosteroids as an adjuvant therapy of bacterial meningitis, the authors found that corticosteroids did not reduce the overall mortality in meningitis patients but can reduce hearing loss and neurological sequelae [145]. The effect of corticosteroids on meningitis mortality and sequelae varied according to the bacterial agent causing meningitis [145]. Benefits of corticosteroids in treatment of meningitis patients have led to hypothesis and experimental approaches to modulate the immune response in order to decrease the harmful effects of inflammation and to improve the outcome of bacterial meningitis [146]. In one study, the benefit of prophylactic palmitoylethanolamide (a natural fatty acid amide) was demonstrated in a mouse model of *E. coli* meningitis to prolong survival and reduce symptoms by reducing inflammation and slowing the progression of infection [147]. Despite success as immunomodulation therapy for a number of auto-immune diseases such as arthritis and psoriasis, this approach, other than the use of dexamethasone, as adjuvant therapy of bacterial meningitis remain elusive and at the pre-clinical stages of development.

## 9. Looking Ahead and What to Expect in the Post-Genomic Era of Meningitis Control

The conjugate vaccines currently in use to control *S. pneumoniae*, *N. meningitidis*, and *H. influenzae* invasive infections have undoubtedly saved tens of thousands of lives [148] but there is no room for complacency because these vaccines do not offer universal coverage against all serotypes or serogroups of these pathogens. Therefore, laboratory surveillance couples with good epidemiological work remain important to monitor the trends of vaccine-preventable bacterial meningitis. We need to stay vigilant for diseases due to strains arising from the phenomenon of capsule switching and capsule replacement. For *H. influenzae*, the most common invasive strains now are non-encapsulated [91]. An increase in the detection of non-encapsulated *S. pneumoniae* has recently been reported [149]. Although most non-encapsulated *S. pneumoniae* do not cause IPD, their increase in prevalence may be concerning since they may serve as reservoirs of gene pools, including antibiotic resistance genes, for transfer into encapsulated *S. pneumoniae*. Another concern is the finding of hybrid capsules [150,151] including new capsule types due to recombination with a different *Streptococcus* species, for example, *S. mitis* [29]. Transfer of a *S. pneumoniae* capsule into a normally non-pathogenic or non-invasive *S. mitis* strain has also been reported [152]. With a large repertoire of capsule genes in *S pneumoniae*, and related *Streptococcus* species, there may be endless combinations for the organisms to take advantage of to evade vaccine immunity.

The genomic era seems to have opened up new opportunities like “reverse vaccinology” to quickly identify potential vaccine candidates [153]. Machine-learning and artificial intelligence have also been proposed to mine genomes for useful data and genes for potential applications [154].

## 10. Conclusions

Nowadays, we have powerful conjugate vaccines that target the most common bacterial meningitis agents (at least the most common invasive serotypes or serogroups) to not only prevent infections in the vulnerable age group, but also by eliminating nasopharyngeal carriage, to provide herd immunity to the non-vaccinated individuals. Conjugate vaccines have prevented millions of deaths from bacterial meningitis over the last two decades [2]. We now also have genomic tools that can read the complete coding sequences of bacteria for a never-before-seen gene-by-gene comparison at the nucleotide sequence level to identify and track the movement of strains (including new strains) and infections globally [76,84,97,133] in order to either quickly deploy vaccines or to develop newer vaccines for control. Nevertheless, we cannot be complacent as we have witnessed changes in the three bacterial meningitis agents after vaccine introduction. The significant increase of invasive *H. influenzae* disease due to non-encapsulated or non-typeable strains or the increase in Hia in some population in recent years are of concern [94,95,96,97,98]. The epidemiology of IMD in Africa has changed with much success in the deployment of the monovalent MenAfriVac leading to dramatic decreases in incidences of serogroup A diseases [81]. However, other vaccine-preventable serorgroups like W and C still continue to cause significant amount of disease when vaccines against these serogroups have not been deployed yet. The most problematic may be related to IPD due to non-vaccine serotypes emerging to cause disease after the sequential introduction of PCV7, PCV10 and PCV13 [50,52,53,57,58]. Whether this is related to the large number of serotypes of *S. pneumoniae* in contrast to the much smaller number of serotypes of *H. influenzae* or serogroups of *N. meningitidis* is unknown, but mathematical modelling suggested the number of serotypes might have an effect on strain replacement in nasopharyngeal carriage after vaccination [49]. Even though only 10 serotypes of *S. agalactiae* have been identified, its different ecology (genito-gastrointestinal colonizer versus pharyngeal colonizer) may make the effect of conjugate vaccines on the subsequent epidemiology difficult to predict.

In summary, we are in a much better position to control bacterial meningitis than ever before and surveillance continues to have a key role to play [69,70].

## Figures and Tables

**Figure 1 microorganisms-09-00449-f001:**
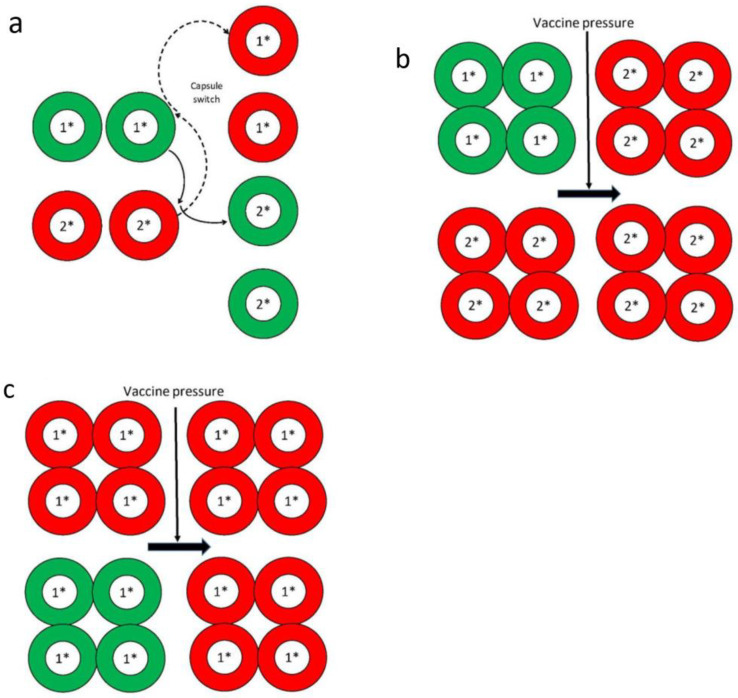
(**a**) Depiction of capsule switching between a genetic lineage 1* strain with a vaccine-type capsule (colored green) and a genetic lineage 2* strain with a non-vaccine type capsule (colored red). Diagram based on information from Swartley et al. [48]. (**b**) Illustration of capsule replacement when the vaccine capsule type (colored green) strain of genetic lineage 1* is removed by the vaccine, leaving the strain of genetic lineage 2* with non-vaccine capsule (colored red) to remain and proliferate. Diagram based on information from Lipstich M [49]. (**c**) Another scenario of capsule replacement when strain of genetic lineage 1* with both vaccine (colored green) and non-vaccine (colored red) capsule types are present before vaccine introduction and after vaccine use, only the non-vaccine capsule type of genetic lineage 1* strain remains. Diagram based on information from Lipstich M [49].

**Table 1 microorganisms-09-00449-t001:** Capsular antigens of *Haemophilus influenzae*, *Neisseria meningitidis*, *Streptococcus agalactiae*, and *S. pneumoniae*.

Organism	Capsular Serotype/Serogroup Antigens	Reference
*H. influenzae*	Serotypes a, b, c, d, e, and f,	Pittman, 1931 [26]
*N. meningitidis*	Serogroups A, B, C, E, H, I, K, L, W, X, Y, and Z	Harrison et al., 2013 [27]
*S. agalactiae*	Serotypes Ia, Ib, II, III, IV, V, VI, VII, VIII, and IX	Song et al., 2018 [5]; Lin et al., 2018 [13]
*S. pneumonia* *	100 serotypes have been identified and only a few are listed here; Serotypes 1, 2, 3, 4, 5, 6A, 6B, 6C, …… 11E, 20B, … 35D, 7D, 10D (complete list can be found in the references provided)	Geno et al., 2015 [28]; Ganaie et al., 2020 [29]

***** 100 different serotypes identified, please see references for full list.

**Table 2 microorganisms-09-00449-t002:** Licensed vaccines * against *Haemophilus influenzae*, *Neisseria meningitidis* and *Streptococcus pneumoniae.*

Origanism	Vaccine Type	Serotype/Serogroup Targets	Protein Carrier ^†^	Year First Licensed
*H. influenzae*	Hib conjugate	b	TT, OMP	1987
*N. meningitidis*	tetravalent polysaccharide	A, C, Y and W	None	1974
*N. meningitidis*	monovalent C conjugate	C	CRM_197_, TT	1999
*N. meningitidis*	monovalent A conjugate	A	TT	2010
*N. meningitidis*	tetravalent conjugate	A, C, Y and W	CRM_197_, DT, TT	2005
*N. meningitidis*	4 component MenB	B	protein base vaccine	2013
*N. meningitidis*	factor H binding protein	B	protein base vaccine	2018
*S. pneumoniae*	PCV7 conjugate	4, 6B, 9V, 14, 18C, 19F, 23F	CRM_197_	2000
*S. pneumoniae*	PCV10 conjugate	1, 4, 5, 6B, 7F, 9V, 14, 18C, 19F, 23F	CRM_197_, TT, DT	2009
*S. pneumoniae*	PCV13 conjugate	1, 3, 4, 5, 6A, 6B, 7F, 9V, 14, 18C, 19A, 19F, 23F	CRM_197_	2011
*S. pneumoniae*	PPV23 plain polysaccharide	1, 2, 3, 4, 5, 6B, 7F, 8, 9N, 9V, 10A, 11A, 12F, 14, 15B, 17F, 18C, 19A, 19F, 20, 22F, 23F, 33F	None	1983

* Except for the 4 component MenB and the factor H binding protein vaccines, all the other vaccines described in this Table are polysaccharide based vaccines. ^†^ protein carriers include TT (tetanus toxoid), DT (diphtheria toxoid), CRM_197_ (mutant diphtheria toxoid), OMP (outer membrane protein of *N. meningitides*).

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
