# Peer review of "A Narrative Review of the Molecular Epidemiology and Laboratory Surveillance of Vaccine Preventable Bacterial Meningitis Agents: *Streptococcus pneumoniae*, *Neisseria meningitidis*, *Haemophilus influenzae* and *Streptococcus agalactiae"

_microorganisms, 2021, doi:10.3390/microorganisms9020449_

Round 1
Reviewer 1 Report
This manuscript is a comprehensive, current and well-written review. The author provides a brief overview of the current vaccines against S. pneumonia, H. influenzae and N. meningitidis. As there is no current vaccine for group B strep (which can also contribute to meningitis), the work is meant to provide an expansive look at the current state of surveillance and what types of disease trends are seen for the other three pathogens. This manuscripts outlines very well what considerations should be taken into account for the group B strep vaccine once one is available. The rationale for and description of procedures of surveillance (with description of some potentially new avenues) were described very well. It is a timely and important review.
Minor suggestions:
Line 15 change parasites to pathogens
Line 22 groups
Line 88 remove highlighting
Line 133 font change in influenzae
Line 136 italics of bacterial names
Line 142 and elsewhere "plain polysaccharide" change to "solely polysaccharide-based"
Section 8 - Please ensure bacterial names are italized in all places
Author Response
please see attached response.

Reviewer 2 Report
A review article well written with an important overview on the vaccinology and vaccine-related epidemiology of bacterial meningitis. Yet, to make this review article complete and comprehensive for the field, there are still important gaps in the scientific content that should be taken into consideration by the authors. Please find below my comments and remarks.
- Title: why S. agalactiae is within brackets?
- Table 1: all serotypes of S. pneumoniae should be listed without ”…”.
- Vaccines and prophylactic approaches to prevent meningitis caused by invasive Escherichia coli and Listeria monocytogenes should also be discussed in the review. Both invasive E. coli and L. monocytogenes are important etiological cause of bacterial meningitis.
- Besides vaccines, one section focused on experimental prophylactic approaches to boost immune response towards the bacterial agents causing meningitis should also be included in the review article.
- Figure 1a. Is the schematic representation of capsule switch an hypothesis of the authors, or is an interpretation based on existing literature? If it correlates to evidence from literature, citations should be inserted in the text. Same for 1b and 1c.
Author Response
please see attached response

Round 2
Reviewer 2 Report
Good work of revision. Only one major remark concerning the correct spelling of all bacterial names, for instance "H. influenzae" instead of "H. influenza". All bacterial names should be correct in the final version for publication.
Author Response
Response to reviewer's comments:
The spelling of all bacterial names have been double checked.